# Peripheral T-Cells, B-Cells, and Monocytes from Multiple Sclerosis Patients Supplemented with High-Dose Vitamin D Show Distinct Changes in Gene Expression Profiles

**DOI:** 10.3390/nu14224737

**Published:** 2022-11-09

**Authors:** Dohyup Kim, Emily E. Witt, Simone Schubert, Elias Sotirchos, Pavan Bhargava, Ellen M. Mowry, Karen Sachs, Biter Bilen, Lawrence Steinman, Avni Awani, Zihuai He, Peter A. Calabresi, Keith Van Haren

**Affiliations:** 1Neurology and Neurological Sciences Department, Stanford University School of Medicine, Stanford, CA 94305, USA; 2Harvard Medical School, Boston, MA 02115, USA; 3Department of Environmental Health and Safety, Stanford University School of Medicine, Stanford, CA 94305, USA; 4Department of Neurology, Johns Hopkins University School of Medicine, Baltimore, MD 21205, USA; 5Next Generation Analytics, Palo Alto, CA 94301, USA; 6Data Science and Engineering Consultant, Mountain View, CA 94041, USA

**Keywords:** vitamin D, multiple sclerosis, microarray, differentially expressed genes, immune cells, T-cells, B-cells, monocytes

## Abstract

Vitamin D is a steroid hormone that has been widely studied as a potential therapy for multiple sclerosis and other inflammatory disorders. Pre-clinical studies have implicated vitamin D in the transcription of thousands of genes, but its influence may vary by cell type. A handful of clinical studies have failed to identify an in vivo gene expression signature when using bulk analysis of all peripheral immune cells. We hypothesized that vitamin D’s gene signature would vary by immune cell type, requiring the analysis of distinct cell types. Multiple sclerosis patients (*n* = 18) were given high-dose vitamin D (10,400 IU/day) for six months as part of a prospective clinical trial (NCT01024777). We collected peripheral blood mononuclear cells from participants at baseline and again after six months of treatment. We used flow cytometry to isolate three immune cell types (CD4^+^ T-cells, CD19^+^ B-cells, CD14^+^ monocytes) for RNA microarray analysis and compared the expression profiles between baseline and six months. We identified distinct sets of differentially expressed genes and enriched pathways between baseline and six months for each cell type. Vitamin D’s in vivo gene expression profile in the immune system likely differs by cell type. Future clinical studies should consider techniques that allow for a similar cell-type resolution.

## 1. Introduction

Vitamin D refers to a group of fat-soluble steroid hormones. The most important form of vitamin D in mammals is vitamin D_3_, or cholecalciferol, which is endogenously synthesized in the lower layers of skin through a chemical reaction regulated by exposure to ultraviolet-B radiation, typically via sunlight. Dietary sources (e.g., dairy, fish, nutritional supplements) constitute an important secondary source. Vitamin D is metabolized via a series of hydroxylation processes in the liver to reach its most stable form, 25-hydroxyvitamin D. The conversion to 1,25-dihydroxyvitamin D is often regulated at the local tissue level and has the highest affinity for the nuclear hormone receptor called vitamin D receptor (VDR) [1]. The ligand-activated transcription factor VDR forms a transcriptional complex by recruiting cofactors and binds to accessible genomic regions to regulate expression levels of over 500 genes in different tissues in vivo including, skin, kidney, intestine, bone, and immune cells [2,3]. As a result, vitamin D plays an important role in diverse physiological functions such as maintaining bone and calcium homeostasis, controlling cellular growth and differentiation, and modulating immunity. Numerous studies showed that the deficiency of vitamin D is closely associated with a wide variety of human diseases, including cancer, diabetes, cardiovascular diseases, infectious diseases, mental illnesses, and autoimmune disorders, although the causal and mechanistic nature of these associations is often poorly defined [4].

Multiple sclerosis (MS), a chronic inflammatory disease of the central nervous system, has been causally linked to vitamin D deficiency via mendelian randomization; however, the nature of the mechanism has not been firmly established [5,6,7,8]. Progression of MS is characterized by blood–brain barrier breakdown, perivascular inflammation, and oligodendrocyte dysfunction, leading to progressive neurodegeneration [9]. While MS is susceptible to genetic factors, there also are multiple environmental risk factors in disease onset, including Epstein–Barr virus infection, smoking, obesity, season of birth, virus infection, and vitamin D deficiency [10,11,12,13,14]. Among them, a low serum level of vitamin D is considered a major environmental factor associated with disease severity and incidence [15]. For example, mothers with high vitamin D levels before and during pregnancy had children with a lower risk of MS [16,17]. Furthermore, during adolescence, intake of vitamin-D-rich food such as fatty fish and cod liver oil significantly decreased the risk of MS [18,19]. Therefore, researchers have investigated vitamin D’s immunomodulating effect as a possible therapeutic agent for MS.

Vitamin D’s role in MS pathogenesis is widely presumed to result from its regulatory effects on immune response and immune cell activation [20]. In vitro studies of vitamin D have amply demonstrated its ability to modify the phenotype and function of various immune cells via the VDR [1]. Among them, T cells play an important role in the inflammation in MS. A high dose of vitamin D reduces the number of interleukin-17 (IL-17)-producing CD4^+^ T-cells and increases central memory CD4^+^ T-cells and naïve CD4^+^ T-cells [15,21]. In a randomized, double-blind trial, MS patients who took high-dose vitamin D for six months showed a reduction in IL-17 expression, which is a significant contributor to the immunopathogenesis of MS [21]. In a mouse model, vitamin D supplementation down-regulated disease-driving pathways in CD4^+^ T cells [22]. On the other hand, monocytes are known to be the most vitamin D-responsive cell type in the immune system [23]. Monocytes cultured with vitamin D supplementation showed a significant reduction in the ability to induce antigen-dependent T-cell proliferation [24]; they decreased the expression of the surface co-stimulatory molecules [25]. There also is increasing evidence that B cells play an important role in MS; however, in vitro results of vitamin D effects on B cells are not supported by in vivo data [26].

Despite abundant in vitro evidence of vitamin D’s effects on immune cell function, its gene expression profiles have proven difficult to study in vivo, including clinical trials [23]. The few studies that have investigated the gene expression profiles from the immune cells of study participants in clinical trials of vitamin D supplementation have failed to identify consistent signatures [27,28,29]. This failure could be due to the result of highly specialized cell-type responses that cannot be resolved on the bulk sequencing of mixed immune cell populations [30]. Similar cell-specific responses have been observed with other hormones such as cortisol [31], growth hormones [32], and thyroid hormones [33].

We hypothesized that the gene expression profiles from vitamin D treatment are difficult to find when looking at bulk RNA-Seq data and that the true effect of vitamin D treatment can only be identified in cell-type-specific manners. We examined the results of the cell-type-specific gene expression profiles of MS patients. We set out to test this hypothesis by analyzing the gene expression signatures of vitamin D supplementation within major immune cell subsets (CD4^+^ T cells, CD14^+^ monocytes, and CD19^+^ B cells) from peripheral blood mononuclear cells (PBMCs) drawn from MS patients enrolled in a six-month trial of high-dose vitamin D supplementation.

## 2. Materials and Methods

### 2.1. Study Design, Sample Collection, and Isolation

The sample collection was conducted as part of a single-center, randomized, double-blind study that was conducted between April 2010 and January 2013 at the Johns Hopkins Hospital. Johns Hopkins University Institutional Review Board approval was obtained for the study protocol, and written informed consent was obtained from all participants. Sample analysis was completed at Stanford University under IRB Protocol 23596. The trial was registered with ClinicalTrials.gov (NCT01024777). Although the full trial randomized 40 patients to receive high (10,000 IU/day) or low (400 IU/day) doses, funding constraints limited our study to specimens from participants enrolled in the high-dose treatment arm. Therefore, whole blood samples from the high-dose treatment group only were collected from participants at baseline and 6 months. Inclusion criteria were a diagnosis of relapsing-remitting MS, age 18–55 years, and screening (within one month of baseline) serum 25-hydroxyvitamin D level of 20–50 ng/mL. Exclusion criteria were the existing use of high-dose vitamin D supplementation (defined as a daily intake 1000 IU or greater) or a change of immunomodulatory therapy within the past 3 months; systemic glucocorticoid therapy or relapse within 30 days; pregnancy; serum creatinine >1.5 mg/dL; hypersensitivity to vitamin D preparations; and a history of hyperparathyroidism, tuberculosis, sarcoidosis, or nephrolithiasis (Table 1). 

In total, 18 participants were randomized to receive 10,000 IU/day of cholecalciferol (Continental Vitamin Company, Vernon, CA, USA) daily for 6 months. PBMCs were isolated from whole blood using density gradient centrifugation with a lymphocyte separation medium. PBMCs were then cryopreserved and stored in liquid nitrogen for further analysis. Isolated human PBMCs were thawed quickly in a 37 °C water bath. Cells were slowly diluted and washed three times with Dulbecco’s phosphate-buffered saline without calcium and magnesium to avoid activating the cells.

### 2.2. Cell Sorting

Cells were sorted by Fluorescent-Activated Cell Sorting (FACS) into three cell populations: CD4^+^ (T cells), CD14^+^ (monocytes), and CD19^+^ (B Cells). Human PBMCs were stained with the following antibodies from BD Biosciences (San Jose, CA, USA): FITC conjugated anti-CD14 (347439#), APC conjugated anti-CD19 (340437#), and PE conjugated anti-CD4 (347327#). All samples were treated with a LIVE/DEAD cell viability assay (Invitrogen, Carlsbad, CA, USA) and gated on cell viability. Two samples with less than 70% cell viability were excluded (*n* = 18). Lymphocytes and monocytes were sorted directly into 24-well plates with 0.75 mL Trizol Reagent (Invitrogen, Carlsbad, CA, USA) to maximize RNA extraction.

### 2.3. RNA Isolation

RNA was extracted from sorted cells using a customized Trizol/Chloroform method. Total RNA was subsequently harvested and purified using RNeasy Mini Elute Kits (Qiagen, Austin, TX, USA) in an RNase-free environment according to the manufacturer’s protocol. The purity and integrity of total RNA were determined by NanoDrop ND-1000 UV-vis spectrophotometer (NanoDrop Technologies Inc., Wilmington, DE, USA) and Agilent 2100 Bioanalyzer (Agilent Technologies, Santa Clara, CA, USA). From all samples submitted for microarray analysis, two samples did not meet the quality threshold of an RNA amount between 5 and 20 ng, with an RNA integrity (RIN) score of at least 7.7 (*n* = 16). Samples were hybridized on Affymetrix Genechip^TM^ Human Gene 2.0 ST arrays (Affymetrix, Santa Clara, CA, USA).

### 2.4. Data Analysis Plan

Our primary objective was to identify differentially expressed genes (DEGs) between baseline and final study visits at 6 months within each of the FACS sorted immune cell groups: CD4^+^ T-cells, C14^+^ monocytes, and CD19^+^ B-cells. We used principal component analysis to visualize overall differences in gene expression profiles between baseline and 6 months, followed by a DEG analysis of each individual gene. Our secondary objective was to identify cell-type-specific genes and molecular pathways affected by the vitamin D supplementation. We also sought to identify shared DEGs and pathways across different immune cell types and compare them against the previous findings in the literature.

### 2.5. Microarray Data Processing and Analysis

Microarray raw data were quality checked with hybridization controls and normalized using Transcriptome Analysis Console (TAC) Software v4.0.2 (Thermo Fisher Scientific, Waltham, MA, USA) according to their manual. To minimize the variation from individual patients, we performed paired comparisons between baseline and six-month data, only using the data of the patients present with both timepoints. This resulted in sample sizes of 13, 13, and 11 for CD4^+^, CD14^+^, and CD19^+^, respectively. Principal component analysis was plotted using R v4.1.3 (R Foundation for Statistical Computing, Vienna, Austria). We combined the three datasets to mimic the effect of bulk RNA-Seq, where all different cell types are combined and sequenced together. When multiple probes point at the same gene, the probe with the smallest *p* value was selected, as suggested in [34]. Differentially expressed genes were calculated using limma v. 3.46.0 [35]. The threshold for DEGs were a *p* value < 0.05 and >1.2-fold change. 

### 2.6. Gene Set Enrichment Analysis

Gene Set Enrichment Analysis (GSEA) was used to detect relevant biological differences that can be lost in the inherent noise of microarray technology by comparing a priori set of genes known to be involved in previously elucidated biological processes [36]. Statistically significant enriched gene sets grouped according to the Gene Ontology biological pathways were determined at nominal *p* values < 0.05 and suggestive enriched gene sets at nominal *p* values < 0.1.

## 3. Results

Among bio-archived PBMCs collected from the 18 participants assigned to the high-dose arm (10,000 IU/day), two samples were excluded after PBMC thaw and FACS separation due to cell viability <70%. After RNA isolation and purification, two additional samples were excluded because they did not meet the RNA quality threshold (RNA amount < 5 ng; RIN score < 7.7). The remaining samples available for analysis were CD4^+^ (*n* = 16 baseline; *n* = 15 6mos), CD14^+^ (*n* = 16 baseline; *n* = 14 6mos), and CD19^+^ (*n* = 14 baseline; *n* = 11 6mos). Among those, we only included the patient samples with both the baseline and six-month data to reduce the individual patients’ variability and perform pairwise comparisons.

We conducted principal component analysis (PCA) to visually investigate how gene expression profiles were distributed between all microarray samples (Figure 1). The gene expression profiles reveal that the most prominent variation comes from the cell types (Figure 1A). Using a multi-response permutation procedure, we statistically confirmed that each cell type clustered tightly together in ordination space away from other cell types (Figure 1A; *p* value < 0.001, A = 0.2897). We then investigated the gene expression profiles within different cell types (CD4^+^, CD14^+^, and CD19^+^). PCA plots for each cell type appear to distinguish between the periods before and at the end of vitamin D supplementation, although those signatures are not as strong as the cell-type distinction (Figure 1B–D).

To investigate which genes have been affected by vitamin D treatment in MS patients, we identified differentially expressed genes (DEGs) before and after vitamin D treatments. Within CD19^+^ cell type, 564 genes were differentially expressed before and after vitamin D treatments (304 genes up-regulated and 260 genes down-regulated after the treatment; >1.2×-fold change, *p* value < 0.05; Figure 2A). Similarly, 200 and 435 genes were differentially expressed within CD14^+^ and CD4^+^ cells, respectively. Only nine DEGs were present in all three cell types (Figure 2B). These genes include ATP-binding cassette subfamily F member 1 (*ABCF1*), cofilin 1 (*CFL1*), FK506-binding protein 11 (*FKBP11*), neurobeachin-like 1 (*NBEAL1*), ribosomal protein L6 (*RPL6*), ribosomal protein S27a (*RPS27A*), small nucleolar RNA, H/ACA Box 31 (*SNORA31*), non-coding small nuclear component of U6 small nuclear ribonucleoprotein (*U6*), and Y RNA.

When all samples were combined, 31 genes were significantly up-regulated and 46 genes were significantly down-regulated after vitamin D treatment (>1.2×-fold, *p* value < 0.05; Figure 2A,C). Four genes belonged to the DEGs from all four groups (CD4^+^, CD14^+^, CD19^+^, and combined; Figure 2B). These genes were *FKBP11*, *NBEAL1*, *RPS27A*, and *Y RNA*. Additionally, five genes were present in all three cell-type-specific samples, but not the combined group (CD4^+^, CD14^+^, and CD19^+^; Figure 2B). Such genes were *ABCF1*, *CFL1*, *RPL6*, *SNORA31*, and *U6*. Interestingly, 21 new genes that did not belong to any cell-type-specific DEGs were identified as differentially expressed in the combined sample group (Figure 2B), indicating that the combined profile of gene expression does not accurately represent any of the three cell types and that it may represent a complete different set of genes.

We also performed the GSEA to investigate which pathways are over-represented under vitamin D treatment using the ranked list of DEGs based on the paired t statistics. The GSEA of the cell-type-specific data against 50 hallmark gene sets from Molecular Signatures Database (MSigDB) indicated that the effect of vitamin D treatment was mainly on the immune system (Table 2). CD4^+^ T cells had four hallmark gene sets that were up-regulated after vitamin D treatment than before the treatment (heme metabolism, allograft rejection, MYC targets V1, and G2M Checkpoint) and one gene set that was significantly down-regulated (KRAS signaling down). CD14^+^ monocytes had one up-regulated gene set (spermatogenesis) and two gene sets significantly down-regulated after vitamin D treatment (interferon gamma response and complement). CD19^+^ B cells had five significantly down-regulated gene sets (MYC targets V1, MTORC1 signaling, TNFA signaling via NFKB, apoptosis, and heme metabolism). Interestingly, the GSEA result from the combined data showed six significantly down-regulated gene sets, three of which were not present in the cell-type-specific results (Table 2).

## 4. Discussion

In this study, we compared the effects of daily, high-dose vitamin D supplementation on the gene expression profiles of CD4^+^ T cells, CD14^+^ monocytes, and CD19^+^ B cells at baseline and again at 6 months among 18 MS patients enrolled in a clinical trial. To our knowledge, this study is the first study to investigate vitamin D’s in vivo gene expression among distinct immune cell populations. Although the genetic signal from the vitamin D treatment was weak, we were able to observe the cell-type-specific distinction between before and after the vitamin D treatment and identify the differentially expressed genes (DEGs) for each cell type (Figure 1).

Interestingly, the DEGs from three different cell types had minimal overlap. However, the combined data significantly reduced the number of DEGs. Moreover, the combined data had 21 new DEGs that were not present in cell-type-specific data, indicating that the accurate signal from vitamin D may significantly be lost when we analyze the transcriptomic data together.

To compare and contrast our findings, we used two previously published in vivo data [27,37]. The first transcriptome data were from a randomized controlled trial where vitamin D was supplemented to non-MS subjects for three to five years [37]. In this study, the authors failed to find any significant differences in gene expression between before and after the high-dose vitamin D treatment. When they analyzed the men and women separately, they found significant effects on gene expression patterns only in female subjects. While the authors reported 51 genes regulated by vitamin D treatment, none of such genes overlapped with our DEGs. 

Another study identified that 702 genes were significantly affected by vitamin D treatment in human PBMCs [29]. However, this study had major limitations; the population size was small (*n* = 5), vitamin D was administered as one large bolus, and its effect was measured only once after 24 h. In another study, vitamin D supplementation in older adults did not show significant differences in their transcriptomes between treatment and placebo groups [28]. Such unclear results from in vivo studies could be due to vitamin D’s different immunomodulating effects on different types of immune cells.

The second transcriptome data were from a clinical trial of daily vitamin D supplementation for non-MS people aged over 65 years for 12 months [27]. Although the authors observed higher plasma levels of 25-hydroxyvitamin D in the subjects treated with vitamin D, they found no significant effect on whole-blood gene expression compared to the placebo group. A comparison of these two transcriptomic data of vitamin D treatment with our data using GSEA did not show coherence. This result indicates that the methods used in previous studies may not be sufficient to identify the true effect of vitamin D supplementation on gene expression.

Although not yet fully understood, vitamin D levels and MS disease progression or prevention are closely related. Vitamin D administration prevents the progression of MS in the mouse model of human MS, i.e., the experimental autoimmune encephalomyelitis mouse model [38]. High levels of circulating vitamin D in the blood are significantly associated with a lower risk of MS [39]. The genome-wide study found that the genetic abnormality in *CYP27A1* and *CYP27B1*—genes that encode the rate-limiting enzyme to convert vitamin D into its active form—increased the risk of developing MS [40]. While we have mounting phenotypic evidence on the MS and vitamin D level, we still lack the genetic mechanism of how vitamin D works to reduce disease progression in MS. Our study is, to our knowledge, the first cell-type-specific in vivo transcriptomic data of MS patients with vitamin D supplementation. Although we did not observe significant genetic signatures of vitamin D treatment in immune cells, we did find that the effect of vitamin D may be very different in different immune cells.

This study is part of a larger clinical study with both high-dose and low-dose vitamin D treatments [21]. As such, there are a few limitations of the study that came from the design of the original study. First, this study does not have a placebo group that could give us a clear cell-type-specific effect from vitamin D treatment. However, our ethics committee did not agree with administering a placebo to the patients with vitamin D deficiency. Second, as seen in Table 1, the patients with different disease-modifying treatments (DMTs) were included in the study, which may cause different biomarkers and molecular mechanisms from vitamin D treatment. However, we chose to include patients with different DMTs since (1) there have been earlier studies showing that the immunological responses from different DMTs are independent of vitamin D supplementation [41,42,43], and (2) the patients in this study had been on a stable dose of DMT for at least 6 months before the trial so that vitamin D levels were responsible for any new changes in genetic signatures.

## 5. Conclusions

Despite that the exact mechanism of the disease progression is yet to be elucidated, there are potentially multiple layers by which vitamin D influences MS. Vitamin D upregulates different anti-inflammatory pathways [44]. Traditional bulk RNA-Seq and microarray techniques are cheap, fast ways to measure the gene expression levels of the given samples; however, it also has a serious limitation of combining all the signals from different types of cells within a collected sample. The human blood sample contains various immune cell types, each of which is also very heterogeneous (reviewed in [45]). Supplementation of vitamin D is known to affect VDR, which in turn regulates the expression levels of numerous genes by acting as a transcription factor. As a result, vitamin D treatments may have a subtle and indirect effect on gene expression levels through VDR, depending on the types of cells. Such an effect can easily be lost when different cell types are combined. Therefore, it is very important to look at the finer resolution to delineate the true effect of vitamin D.

## Figures and Tables

**Figure 1 nutrients-14-04737-f001:**
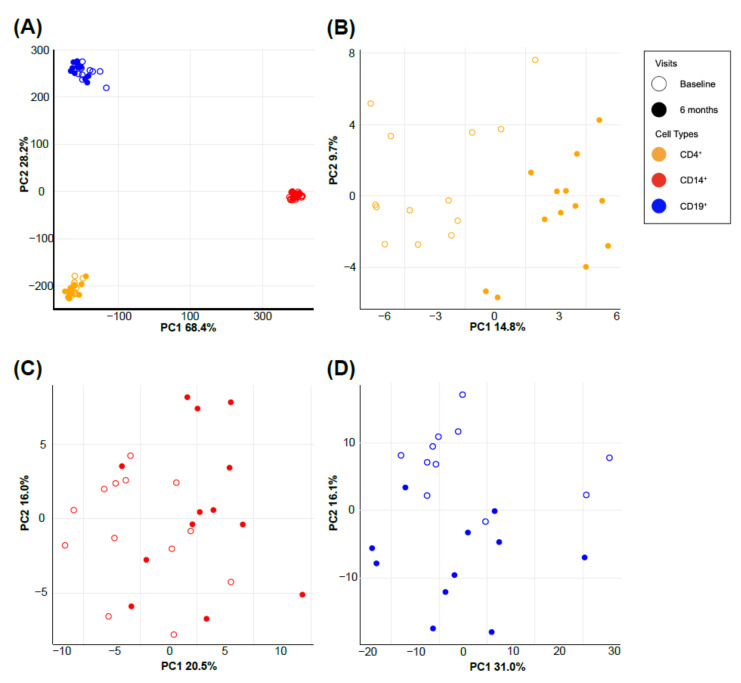
Gene expression profiles show distinct grouping by cell types on the principal component analysis plot. The largest source of variation from PCA is cell type (**A**). PCA plots of cell-type-specific gene expression profiles show distinctions between the periods before (baseline visit) and at the end of (6-month visit) vitamin D supplementation in CD4^+^ T cells (**B**), CD14^+^ monocytes (**C**), and CD19^+^ B cells (**D**). PCA, principal component analysis.

**Figure 2 nutrients-14-04737-f002:**
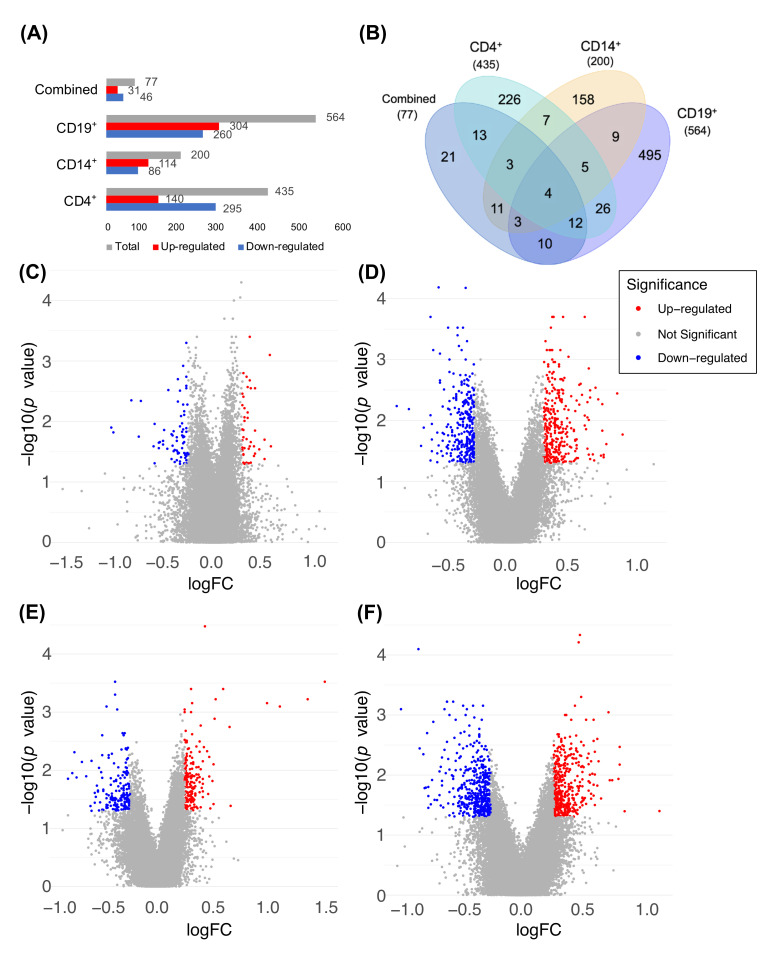
Differentially expressed genes (DEGs) are different between each cell type. More DEGs are down-regulated after vitamin D treatment than DEGs up-regulated in CD4^+^, while combined CD14^+^ and CD19^+^ have more up-regulated DEGs than down-regulated ones (**A**). The majority of DEGs are cell-type-specific, while only four genes are shared between CD4^+^, CD14^+^, CD19^+^, and combined (all cell types combined) (**B**). Nominal *p* value < 0.05, fold change of >1.2×. The combined dataset (**C**) has a much smaller number of DEGs when compared to cell-type-specific CD4^+^ (**D**), CD14^+^(**E**), and CD19^+^ (**F**) data.

**Table 1 nutrients-14-04737-t001:** Patient demographic.

Patient Demographics from MS Cohort	
Age, year, mean (SD)	41.3 (8.1)
Female, *n*, (%)	13 (72)
Race, *n*, (%)	
Caucasian	14 (78)
African American	4 (22)
Immunomodulatory therapy, *n*, (%)	
Interferon-beta	5 (28)
Glatiramer acetate	3 (17)
Natalizumab	6 (33)
Fingolimod	2 (11)
Untreated	2 (11)
Serum 25-hydroxyvitamin D at baseline (ng/mL), mean (SD)	24.51 (2.13)
Serum 25-hydroxyvitamin D at six months (ng/mL), mean (SD)	61.72 (4.04)

MS, multiple sclerosis; SD, standard deviation.

**Table 2 nutrients-14-04737-t002:** Significantly up- or down-regulated gene sets from gene set enrichment analysis (nominal *p* value < 0.1).

	Name	Size ^1^	NES ^2^	Nominal *p* Value	FDR *q* Value ^3^
CD4^+^	Up-regulated after vitamin D treatment				
	**Heme Metabolism**	6	0.537	**0.034**	0.199
	**Allograft Rejection**	6	0.548	**0.034**	0.33
	**MYC Targets V1**	14	0.383	**0.042**	0.56
	G2M Checkpoint	9	0.413	0.076	0.507
	Down-regulated after vitamin D treatment				
	KRAS Signaling Down	7	−0.491	0.056	0.260
CD14^+^	Up-regulated after vitamin D treatment				
	Spermatogenesis	6	0.520	0.061	0.240
	Down-regulated after vitamin D treatment				
	**Interferon Gamma Response**	10	−0.73	**<0.001**	0.015
	**Complement**	5	−0.711	**0.009**	0.012
CD19^+^	Up-regulated after vitamin D treatment				
	None				
	Down-regulated after vitamin D treatment				
	**MYC Targets V1**	9	−0.600	**<0.001**	0.138
	**MTORC1 Signaling**	6	−0.586	**0.022**	0.084
	TNFA Signaling via NFKB	7	−0.480	0.060	0.182
	Apoptosis	5	−0.517	0.085	0.145
	Heme Metabolism	9	−0.405	0.098	0.297
Combined	Up-regulated after vitamin D treatment				
	KRAS Signaling Down	10	0.449	0.051	0.814
	Down-regulated after vitamin D treatment				
	**E2F Targets**	8	−0.584	**0.007**	0.052
	**Allograft Rejection**	7	−0.601	**0.013**	0.081
	**Interferon Gamma Response**	5	−0.618	**0.031**	0.121
	**Xenobiotic Metabolism**	8	−0.471	**0.036**	0.158
	**Protein Secretion**	5	−0.592	**0.039**	0.060
	Complement	5	−0.564	0.075	0.058

^1^ Number of genes that belong to the gene set; ^2^ normalized enrichment score; ^3^ false discovery rate-corrected *q* value. MYC, *MYC* proto-oncogene; G2M, cell cycle G2/M phase; KRAS, Kirsten rat sarcoma virus gene; MTORC1, mechanistic target of rapamycin complex 1; TNFA, tumor necrosis factor A; NFKB, nuclear factor kappa-light-chain-enhancer of activated B cells; E2F, E2 factor; pathways and *p* values in bold have nominal *p* values < 0.05.

## Data Availability

Data presented in this study are available on request from the corresponding author.

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
