# Peer review of "Peripheral T-Cells, B-Cells, and Monocytes from Multiple Sclerosis Patients Supplemented with High-Dose Vitamin D Show Distinct Changes in Gene Expression Profiles"

_nutrients, 2022, doi:10.3390/nu14224737_

Round 1

Reviewer 1 Report

This study investigated the effects of high-dose vitamin D on different peripheral mononuclear cells (CD4+T lymphocyte, CD14+ monocyte, CD19+ B lymphocyte) in MS patients. The study is valuable and original because it examines the in vivo effect of vitamin D on immune cells. Down-regulating effects of high-dose vitamin D on interferon gamma response and TNFA signaling via NFKB may be associated with anti-inflammatory effects and positive effects on MS.

I have only one question regarding the study. Were blood samples collected from subjects receiving low-dose vitamin D? Did you compare the high-dose vitamin D group and the low-dose group?

On the other hand, there is an error in the number of patients. While the number of patients is 19 in the material-method, and results sections, the number of patients in the discussion section is 18.

Author Response

Point 1: I have only one question regarding the study. Were blood samples collected from subjects receiving low-dose vitamin D? Did you compare the high-dose vitamin D group and the low-dose group?

Response 1: We apologize the confusion from the manuscript not being clear. The blood samples were collected from subjects with high-dose vitamin D as part of a larger clinical trial. Due to limited funding and resources, we limited the scope of our analysis to the high-dose arm of the study and completed RNA microarray analysis only on the PBMC samples from these participants. While our original methods included language describing this limitation, we have added language to further emphasize and clarify this aspect of our approach and the underlying rationale. The revised portion of our methods now reads as follows (lines 121-125):

“Although the full trial randomized 40 patients to receive high (10,000 IU/day) or low (400 IU/day) doses, funding constraints limited our study to specimens from participants enrolled in the high dose treatment arm. Therefore, whole blood samples from high-dose treatment group only were collected from participants at baseline and 6 months.”

Point 2: On the other hand, there is an error in the number of patients. While the number of patients is 19 in the material-method, and results sections, the number of patients in the discussion section is 18.

Response 2: We thank the reviewer to point out the error in the manuscript. The number of patients has been corrected to 18 in the methods and results sections (lines 37, 134, and 191)

Reviewer 2 Report

I reviewed the manuscript entitled "Peripheral T-cells, B-cells, and monocytes from multiple sclerosis patients supplemented with high-dose vitamin D show distinct changes in gene expression profiles". It's a novel and well-written paper which showed the impacts of vitamin D supplementation on MS inflammatory markers. However, there are major limitations, as the authors have mentioned. To be added to the limitations, the lack of a placebo group, possible different patterns of vitamin D and biomarkers changes due to DMTs and baseline vitamin D, and controlling the impacts of DMTs on the vitamin D changes. They have to be added and discussed in the discussion. They could all be summarized and recommended in a conclusion for future studies. 

Author Response

Point 1: I reviewed the manuscript entitled "Peripheral T-cells, B-cells, and monocytes from multiple sclerosis patients supplemented with high-dose vitamin D show distinct changes in gene expression profiles". It's a novel and well-written paper which showed the impacts of vitamin D supplementation on MS inflammatory markers. However, there are major limitations, as the authors have mentioned. To be added to the limitations, the lack of a placebo group, possible different patterns of vitamin D and biomarkers changes due to DMTs and baseline vitamin D, and controlling the impacts of DMTs on the vitamin D changes. They have to be added and discussed in the discussion. They could all be summarized and recommended in a conclusion for future studies. 

Response 1: We thank the reviewer for the helpful feedback. We understand the limitations of the study and have added them right before the last paragraph of the discussion section (lines 311-322). In short, this study is a part of larger clinical study so that the mentioned limitations came from the design of the original study. The new paragraph now reads as follows:

“This study is a part of larger clinical study with both high-dose and low-dose vitamin D treatment [21]. As such, there exist a few limitations of the study that came from the design of the original study. First, this study does not have placebo group that could give us a clear cell-type specific effect from vitamin D treatment. However, our ethics committee did not agree to administer placebo to the patients with vitamin D deficiency. Second, as seen in Table 1, the patients with different disease modifying treatments (DMTs) were included in the study, which may cause different biomarkers and molecular mechanisms from vitamin D treatment. However, we chose to include patients with different DMTs since 1) there have been earlier studies that the immunological responses from different DMTs are independent of vitamin D supplementation [41-43], and 2) the patients in this study had been on stable dose of DMT for at least 6 months before the trial so that vitamin D levels are responsible for any new changes in genetic signatures.”